# AnimationPak: Packing Elements with Scripted Animations

Reza Adhitya Saputra*
University of Waterloo

Craig S. Kaplan†
University of Waterloo

Paul Asente‡
Adobe Research

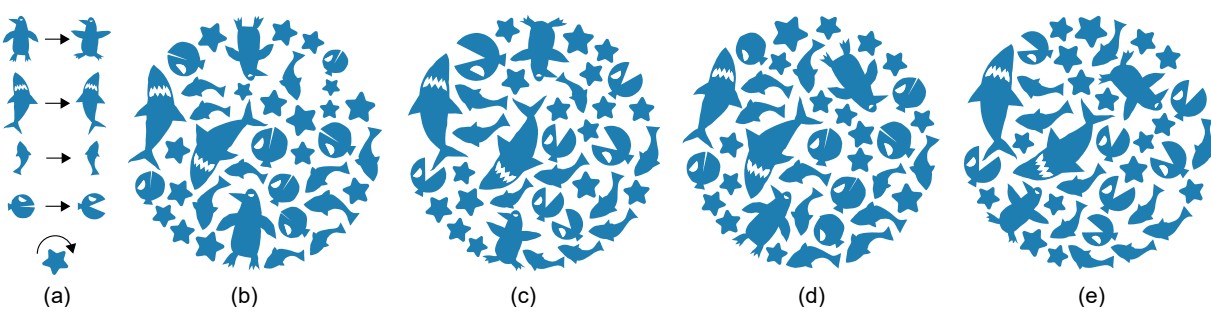

Figure 1: (a) Input animated elements, each with its own animation: swimming penguins, swimming sharks and fish, Pac-Man fish that open or close their mouths, and rotating stars. (b-e) four selected frames from an animated packing.

## ABSTRACT

We present AnimationPak, a technique to create animated packings by arranging animated two-dimensional elements inside a static container. We represent animated elements in a three-dimensional spacetime domain, and view the animated packing problem as a three-dimensional packing in that domain. Every element is represented as a discretized spacetime mesh. In a physical simulation, meshes grow and repel each other, consuming the negative space in the container. The final animation frames are cross sections of the three-dimensional packing at a sequence of time values. The simulation trades off between the evenness of the negative space in the container, the temporal coherence of the animation, and the deformations of the elements. Elements can be guided around the container and the entire animation can be closed into a loop.

**Index Terms:** I.3.3 [Computing Methodologies]: Computer Graphics—Picture/Image Generation; I.3.m [Computing Methodologies]: Computer Graphics—Animation;

## 1 INTRODUCTION

A decorative packing is a composition created by arranging two-dimensional shapes called *elements* within a larger region called a *container*. Packings are popular in graphic design, and are used frequently in advertising and product packaging.

At a high level, packings can communicate a relationship between a whole and the parts that make it up. Consider for example the logo of the 2018 SIGGRAPH conference, shown inset. The 2018 logo surrounds the main logo of the SIGGRAPH organization with a ring of small icons depicting computer graphics themes. 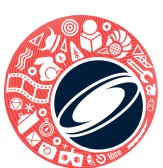

At a lower level, packings must be attractive compositions, which balance the shapes of the elements with the empty space between them, known as the *negative space*. In particular, negative space

*e-mail: radhitya@uwaterloo.ca
†e-mail: csk@uwaterloo.ca
‡e-mail: asente@adobe.com

should be distributed as evenly as possible, leading to roughly constant-width "grout" between elements.

Recently, Saputra et al. presented RepulsionPak [31], a deformation-driven packing method inspired by physical simulation techniques. In RepulsionPak, small elements are placed within a fixed container shape. As they grow, they interact with each other and the container boundary, inducing forces that translate, rotate, and deform elements. The motion and deformation of the elements allows them to achieve a physical equilibrium with an even distribution of negative space.

Inspired by RepulsionPak, we investigate a physics-based packing method for elements with scripted animations. An element can have an animated deformation, such as a bird flapping its wings or a fish flicking its tail. It can also have an animated transformation, giving a changing position, size, and orientation within the container. Our goal is producing an *animated packing*, with elements playing out their animations while simultaneously filling the container shape evenly. A successful animated packing should balance among the evenness of the negative space, the preservation of element shapes, and the comprehensibility of their scripted animations.

In our technique, called *AnimationPak*, we consider an animated element to be a geometric extrusion along a time axis, a three-dimensional object that we call a "spacetime element". We use a three-dimensional physical simulation similar to RepulsionPak to pack spacetime elements into a volume created by extruding a static container shape. The animated packing emerges from this three-dimensional volume by rendering cross sections perpendicular to the time axis. Our time axis behaves differently than a third spatial dimension. Although the cross sections of a spacetime element can drift from their original positions on the time axis, they must remain ordered monotonically. Furthermore, each individual cross section must remain flat in time, so that all of its 2D points occur simultaneously.

Animated packings are a largely unexplored style of motion graphics, presumably because of the difficulty of creating an animated packing by hand. We were not able to find any motivating examples created by artists. There is also very little past research on animated packings; we discuss the work that does exist in the next section.

## 2 RELATED WORK

**Packings and mosaics:** Researchers have explored many approaches to creating 2D packings and simulated mosaics, including using Centroidal Area Voronoi Diagrams (CAVDs) to position elements [15, 16, 33], spectral approaches to create even negative

space [10], energy minimization [23], and shape descriptors [25]. Several approaches have been proposed to extend 2D packing methods to adapt to the challenges of placing them on the surfaces of 3D objects. [6, 7, 19, 37].

Approaches that work with a smaller library of elements but allow them to deform are particularly relevant to AnimationPak. Xu and Kaplan [36] and Zou et al. [38] developed packing methods that construct calligrams inside containers by allowing significant deformation of letterforms. Saputra et al. presented FLOWPAK [32], which deformed long, thin elements along user-defined vector fields. RepulsionPak [30, 31] deformed elements using mass-spring systems and repulsion forces to create compatibilities between element boundaries.

**Animated packings and tilings:** Animosaics by Smith et al. [33] constructed animations in which static elements without scripted animations follow the motion of an animated container. Elements are placed using CAVDs, and advected frame-to-frame using a choice of methods motivated by Gestalt grouping principles. As the container's area changes, elements are added and removed as needed, while attempting to maximize overall temporal coherence. Dalal et al. [10] showed how the spectral approach they introduced for 2D packings could be extended to pack animated elements in a static container. Like us, they recast the problem in terms of three-dimensional spacetime; they compute optimal element placement using discrete samples over time and orientation. However, their spacetime elements have fixed shapes and are made to fit together using only translation and rotation, limiting their ability to consume the container's negative space.

Liu and Veksler created animated decorative mosaics from video input [26]. Their technique combines vision-based motion segmentation with a packing step similar to Animosaics. Kang et al. [21] extracted edges from video and then oriented rectangular tesserae relative to edge directions.

Kaplan [22] explored animations of simple tilings of the plane from copies of a single shape. Elements in a tiling fit together by construction, and therefore always consume all the negative space in the animation.

**3D packings:** AnimationPak fills a 3D container with 3D elements, and is therefore related to other work on constructing freeform 3D packings. Gal et al. [13] presented a method for constructing 3D collages reminiscent of portrait paintings by Arcimboldo. They filled a 3D container with overlapping 3D elements using a greedy approach and a partial shape matching algorithm. Marco [1] decomposed a 3D model into parts that pack tightly into a small build volume, allowing it to be 3D printed with less waste material and packed into a smaller box. Ma et al. [28] developed a heuristic method to create 3D packings that are overlap free. Other work has experimented with example-based packing of 3D volumes [27], or optimized placement based on user interaction [18].

**Derived animations:** AnimationPak falls into the category of systems that create a derived animation based on some input animation. This problem, which requires preserving the visual character of the input, is a longstanding one in computer graphics research. Spacetime constraints [9, 34] allow an animator to specify an object's constraints and goals, and then calculates the object's trajectory via spacetime optimization. Motion warping [35] is a method that deforms an existing motion curve to meet user-specified constraints. Gleicher [14] developed a motion path editing method that allows user to modify the traveling path of a walking character. Bruderlin and Williams [4] used signal processing techniques to modify motion curves. Carra et al. [5] presented a timeslice grammar to procedurally animate a large number of objects.

Previous work has also investigated geometric deformation of animations. Edmond et al. [17] encoded spatial joint relationships using tetrahedral meshes, and applied as-rigid-as-possible shape deformation to the mesh to retarget animation to new characters.

Choi et al. [8] developed a method to deform character motion to allow characters to navigate tight passages. Masaki [29] developed a motion editing tool that deformed 3D lattice proxies of a character's joints. Dalstein et al. [11] presented a data structure to animate vector graphics with complex topological changes. Kim et al. [24] explored a packing algorithm to avoid collisions in a crowd of moving characters. They defined a motion patch containing temporal trajectories of interacting characters, and arranged deformed patches to prevent collisions between characters.

## 3 ANIMATED ELEMENTS

The input to AnimationPak is a library of animated elements and a fixed container shape. AnimationPak currently supports two kinds of animation: the user can animate the shape of each individual element and can also give elements trajectories that animate their position within the container. This section explains how we animate the element shapes using as-rigid-as-possible deformation, and then construct spacetime-extruded objects that form the basis of our packing algorithm. These elements animate "in place": they change shape without translating. The next section describes how these elements can be given transformation trajectories within the container. Size and orientation of an element can be animated either way; they can be specified as an animation of the element's shape, or they can be part of the transformation trajectory.

### 3.1 Spacetime Extrusion

Each element begins life as a static shape defined using vector paths. Following RepulsionPak, we construct a discrete geometric proxy of the element that will interact with other proxies in a physical simulation. The construction of this proxy for a single shape is shown in Fig. 2, and the individual steps are explained in greater detail below.

In order to produce a packing with an even distribution of negative space, we first offset the shape's paths by a distance $\Delta s$, leaving the shape surrounded by a channel of negative space (Fig. 2a). In our system we scale the shape to fit a unit square and set $\Delta s = 0.04$.

Next, we place evenly-spaced samples around the outer boundary of the offset path and construct a Delaunay triangulation of the samples (Fig. 2b). As in RepulsionPak, we will later treat the edges of the triangulation as springs, allowing the element to deform in response to forces in the simulation. We also follow RepulsionPak by adding extra edges to prevent folding or self-overlaps during simulation (Fig. 2c). First, if two triangles $ABC$ and $BCD$ share edge $BC$, then we add a *shear edge* connecting $A$ and $D$. Second, we triangulate the negative space inside the convex hull of the original Delaunay triangulation, and create new *negative space edges* corresponding to the newly created triangulation edges. These negative space edges are used exclusively for internal bracing. The element's concavities can still be occupied by its neighbours.

We refer to the augmented triangulation shown in Fig. 2c as a *slice*. The entire spacetime packing process operates on slices. However, we will eventually need to compute deformed copies of the element's original vector paths when rendering a final animation (Sect. 6). To that end, we re-express all path information relative to the slice triangulation: every path control point is represented using barycentric coordinates within one triangle.

To extend the element into the time dimension, we now position evenly-spaced copies of the slice along the time axis. Assuming that the animation will run over the time interval $[0, 1]$, we choose a number of slices $n_s$ and place slices $\{s_1, \ldots, s_{n_s}\}$, with slice $s_i$ being placed at time $(i - 1)/(n_s - 1)$. Higher temporal resolution will produce a smoother final animation at the expense of more computation. In our examples, we set $n_s = 100$. Fig. 2d shows a set of time slices, with $n_s = 5$ for visualization purposes.

To complete the construction of a spacetime element without animation, we stitch the slices together into a single 3D object. Let

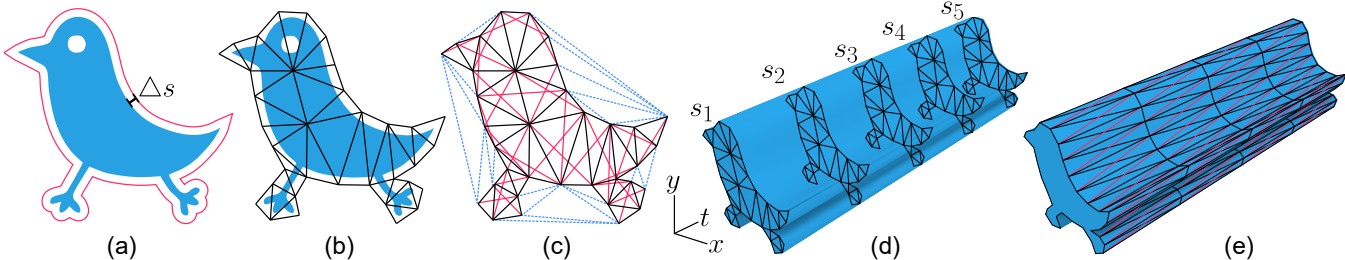

Figure 2: The creation of a discretized spacetime element. (a) A 2D element shape offset by $\Delta s$. (b) A single triangle mesh slice. (c) Shear edges (red) and negative space edges (dashed blue). (d) A set of five slices placed along the time axis. (e) The vertices on the boundaries of the slices are joined by time edges. The black edges in (e) define a triangle mesh called the envelope of the element. In practice we use a larger number of slices in (d) and (e).

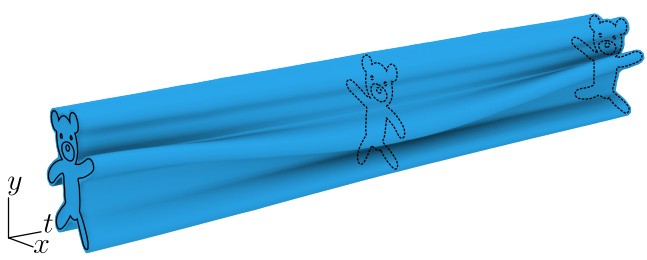

Figure 3: A spacetime element with a scripted animation.

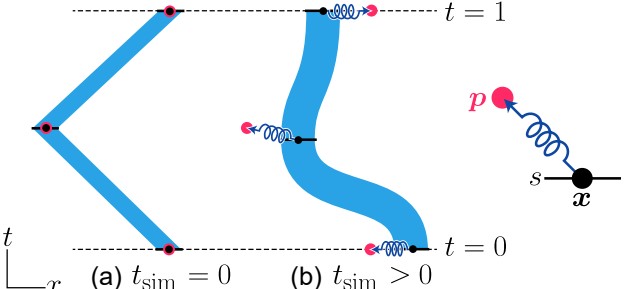

Figure 4: A 2D illustration of a guided element. Slices are depicted as black lines and slice vertices as black dots. A spring connects the centermost vertex $x$ of a slice $s$ to a target point $p$. (a) The initial shape of a guided element is a polygonal extrusion. (b) The spacetime element deforms but the springs pull it back towards the target points.

$s_j$ and $s_{j+1}$ be consecutive slices constructed above. The outer boundaries of the element triangulations are congruent polygons offset in the time axis. We stitch the two polygons together using a new set of *time edges*: if $AB$ is an edge on the boundary of $s_j$ and $CD$ is the corresponding edge on the boundary of $s_{j+1}$, then we add time edges $AC$, $AD$, and $BC$. During simulation, time edges will transmit forces backwards and forwards in time, maintaining temporal coherence by smoothing out deformation and transformations. Fig. 2e shows time edges for $n_s = 5$.

### 3.2 Animation

The 3D information constructed above is a parallel extrusion of a slice along the time axis, representing a shape with no scripted animation. We created a simple interactive application for adding animation to spacetime elements, inspired by as-rigid-as-possible shape manipulation [20]. The artist first designates a subset of the slices as keyframes. They can then interactively manipulate any triangulation vertex of a keyframe slice. Any vertex that has been positioned manually has its entire trajectory through the animation computed using spline interpolation. Then, at any other slice, the positions of all other vertices can be interpolated using the as-rigid-as-possible technique. The result is a smoothly animated spacetime volume like the one visualized in Fig. 3.

Unlike data-driven packing methods like PAD [25], methods that allow distortions do not require a large library of distinct elements to generate successful packings. The results in this paper all use fewer than ten input elements, and some use only one. The physical simulation induces deformation to enhance the compatibility of nearby shapes in the final animation.

### 4 INITIAL CONFIGURATION

We begin the packing process by constructing a 3D spacetime volume for the container by extruding its static shape in the time direction. The container is permitted to have internal holes, which are also extruded. The resulting volume is scaled to fit a unit cube. We also shrink each of the spacetime elements, in the spatial dimensions

only, to 5–10% of its original size. These shrunken elements are thin enough that we can place them in the container without overlaps.

The artist can optionally specify trajectories for a subset of the elements, which we call *guided elements*. A guided element attempts to pass through a sequence of fixed target points in the container, imbuing the animation with a degree of intention and narrative structure. To define a guided element, we designate the triangulation vertex closest to its centroid to be the anchor point for the element. The artist then chooses a set of spacetime target points $\boldsymbol{p}_1, \ldots, \boldsymbol{p}_n$, with $\boldsymbol{p}_i = (x_i, y_i, t_i)$, that the anchor should pass through during the animation. In our interface, the artist uses a slider to choose the time $t_i$ for a target point, and clicks in the container to specify the spatial position $(x_i, y_i)$. The artist can also optionally specify scale and orientation at the target points. We require $t_1 = 0$ and $t_n = 1$, fixing the initial and final positions of the guided element. We then linearly interpolate the anchor position for each slice based on the target points, and translate the slice so that its anchor lies at the desired position. The red extrusions in Fig. 5a are guided elements.

If the artist wishes to create a looping animation, the $(x_i, y_i)$ position for target points $\boldsymbol{p}_1$ and $\boldsymbol{p}_n$ must match up, either for a single guided element or across elements. In Fig. 5 the two guided elements form a connected loop; $(x_1, y_1)$ for each one matches $(x_n, y_n)$ for the other.

In this initial configuration, the guided elements abruptly change direction at target points. However, because the slices are connected by springs, the trajectories will smooth out as the simulation runs. Also, the simulation is not constrained to reach each target position exactly. Instead, we attach the anchor to the target using a *target-point spring* that attempts to draw the element towards it while

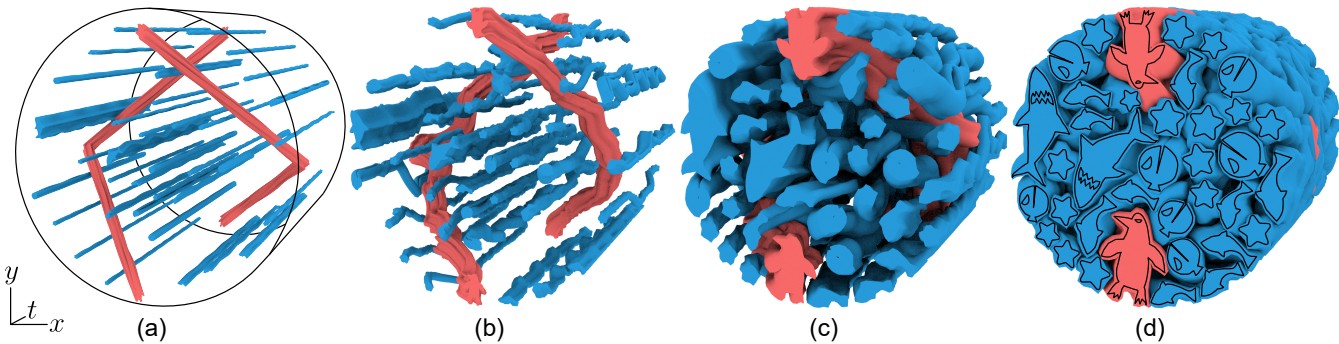

Figure 5: The simulation process. (a) Initial placement of shrunken spacetime elements inside a static 2D disc, extruded into a cylindrical spacetime domain. Guided elements are shown in red and unguided elements in blue. (b) A physics simulation causes the spacetime elements to bend. They also grow gradually. (c) The spacetime elements occupy the container space. (d) The simulation stops when elements do not have sufficient negative space in which to grow, or have reached their target sizes.

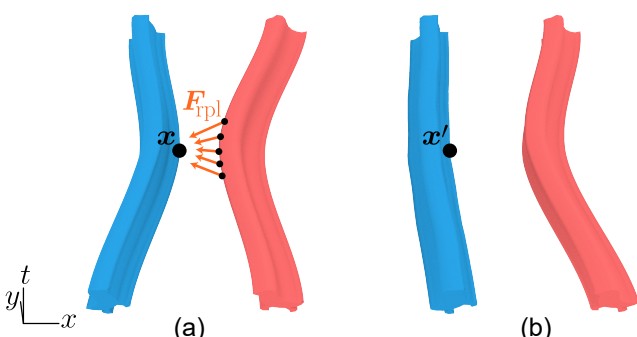

Figure 6: Repulsion forces applied to a vertex $x$, allowing the element to deform and move away from a neighbouring element.

balancing against the other physical forces in play (Fig. 5b). The strength of these springs determines how closely the element will follow the trajectory.

We then seed the container with an initial packing of non-guided spacetime elements. We generate points within the container at random, using blue-noise sampling [2] to prevent points from being too close together, and assign a spacetime element to each seed point, selecting elements randomly from the input library. Depending upon the desired effect, we either randomize their orientations or give them preferred orientations. We reject any candidate seed point that would cause an unguided element's volume to intersect a guided element's volume.

Finally we shrink each element, guided and unguided, uniformly in the spatial dimension towards its centroid. These shrunken elements are guaranteed not to intersect one another; as the simulation runs, they will grow and consume the container's negative space, while avoiding collisions. The blue extrusions in Fig. 5a show an initial placement of spacetime elements.

## 5  SIMULATION

We now perform a physics simulation on the spacetime elements and the container. Elements are subject to a number of forces that cause them to simultaneously grow, deform, and repel each other (Fig. 5). Our physics simulation is very similar to that of RepulsionPak [30] — with the exception of the new temporal force, all our forces are the spacetime analogues of the ones used there. In Sect. 5.2 we introduce some new hard constraints that must be applied after every time step.

Note that we must distinguish two notions of time in this simulation. We use $t$ to refer to the time axis of our spacetime volume,

which will become the time dimension of the final animation, and $t_{\text{sim}}$ to refer to the time domain in which the simulation is taking place.

**Repulsion Forces** allow elements to push away vertices of neighbouring elements, inducing deformations and transformations that lead to an even distribution of elements within the container (Fig. 6). We compute the repulsion force $\boldsymbol{F}_{\text{rpl}}$ on a vertex $\boldsymbol{x}$ located on a slice boundary as:

$$\boldsymbol{F}_{\text{rpl}} = k_{\text{rpl}} \sum_{i=1}^{n} \frac{\boldsymbol{u}}{\|\boldsymbol{u}\|} \frac{1}{\epsilon + \|\boldsymbol{u}\|^2} \tag{1}$$

where
  $k_{\text{rpl}}$ is the relative strength of $\boldsymbol{F}_{\text{rpl}}$. We set $k_{\text{rpl}} = 10$;
  $n$ is the number of nearest points to $\boldsymbol{x}$;
  $\boldsymbol{x_i}$ is the $i$-th closest point on the neighboring element surfaces;
  $\boldsymbol{u} = \boldsymbol{x} - \boldsymbol{x_i}$; and
  $\epsilon$ is a *soft parameter* to avoid instability when $\|\boldsymbol{u}\|$ is small. We set $\epsilon = 1$.

Since the simulation operates in the spacetime domain, vertex $\boldsymbol{x}$ accumulates repulsion forces from points at various time positions. To locate these points on neighbouring elements that are considered nearest, we use a collision grid data structure, described in greater detail in Sect. 5.1.

**Edge Forces** allow elements to deform in response to repulsion forces. The edges defined in Sect. 3 are used here as springs. Like RepulsionPak, we use a non-physical quadratic spring force. Let $\boldsymbol{x_a}$ and $\boldsymbol{x_b}$ be vertices connected by a spring. Each vertex experiences an edge force $\boldsymbol{F}_{\text{edg}}$ of

$$\boldsymbol{F}_{\text{edg}} = k_{\text{edg}} \frac{\boldsymbol{u}}{\|\boldsymbol{u}\|} s \left( \|\boldsymbol{u}\| - \ell \right)^2 \tag{2}$$

where
  $k_{\text{edg}}$ is is the relative strength of $\boldsymbol{F}_{\text{edg}}$. Different classes of spring will have different $k_{\text{edg}}$ values;
  $\boldsymbol{u} = \boldsymbol{x_b} - \boldsymbol{x_a}$;
  $\ell$ is the rest length of the spring; and
  $s$ is +1 or -1, according to whether $(\|\boldsymbol{u}\| - \ell)$ is positive or negative.

We have five types of springs, with stiffness constants that can be set independently. In our implementation we set $k_{\text{edg}}$ to 0.01 for time springs, 0.1 for negative-space springs, and 10 for edge springs, shear springs, and target point springs.

**Overlap forces** resolve a vertex penetrating a neighboring spacetime element. Overlaps can occur later in the simulation when

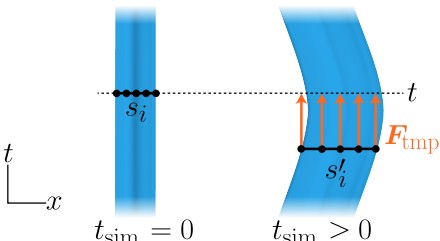

Figure 7: An illustration of the temporal force. The vertices in slice $s_i$ are drawn back towards time $t$.

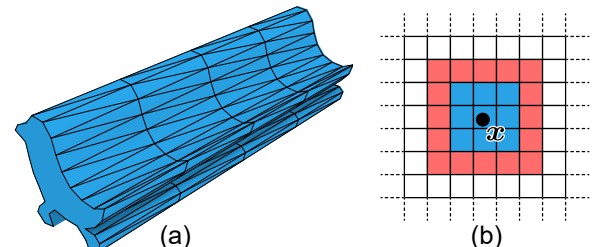

(a)          (b)

Figure 8: (a) The triangles that connect consecutive slices define the envelope of the element. The midpoints of these triangles are stored in a collision grid. (b) A 2D visualization of the region of collision grid cells around a query point $x$ in which repulsion and overlap forces will be computed. In the central blue region, we check overlaps and compute exact repulsion forces relative to closest points on triangles of neighbouring elements; in the peripheral red region we do not compute overlaps, and repulsion forces are approximated using triangle midpoints only.

negative space is limited. Once we detect a penetration, we temporarily disable the repulsion force on vertex $x$, and apply an overlap force $\boldsymbol{F}_{\mathrm{ovr}}$ to push it out:

$$\boldsymbol{F}_{\mathrm{ovr}} = k_{\mathrm{ovr}} \sum_{i=1}^{n} (\boldsymbol{p_i} - \boldsymbol{x}) \qquad (3)$$

where
$k_{\mathrm{ovr}}$ is the relative strength of $\boldsymbol{F}_{\mathrm{ovr}}$. We set $k_{\mathrm{ovr}} = 5$;
$n$ is the number of slice triangles that have $x$ as a vertex; and
$\boldsymbol{p_i}$ is the centroid of the $i$-th slice triangle incident on $x$.

**Boundary forces** keep vertices inside the container. If an element vertex $x$ is outside the container, the boundary force $\boldsymbol{F}_{\mathrm{bdr}}$ moves it towards the closest point on the container's boundary by an amount proportional to the distance to the boundary:

$$\boldsymbol{F}_{\mathrm{bdr}} = k_{\mathrm{bdr}} (\boldsymbol{p_b} - \boldsymbol{x}) \qquad (4)$$

where
$k_{\mathrm{bdr}}$ is the relative strength of $\boldsymbol{F}_{\mathrm{bdr}}$. We set $k_{\mathrm{bdr}} = 5$; and
$\boldsymbol{p_b}$ is the closest point on the target container to $x$.

**Torsional forces** allow an element's slices to be given preferred orientations, to which they attempt to return. Consider a vertex $x$ of a slice, and let $\boldsymbol{c}_r$ be the slice's center of mass in its undeformed state. We define the *rest orientation* of $x$ as the orientation of the vector $\boldsymbol{u}_r = \boldsymbol{x} - \boldsymbol{c}_r$. During simulation we compute the current centre of mass $\boldsymbol{c}$ of the slice and let $\boldsymbol{u} = \boldsymbol{x} - \boldsymbol{c}$. Then the torsional force $\boldsymbol{F}_{\mathrm{tor}}$ is

$$\boldsymbol{F}_{\mathrm{tor}} = \begin{cases} k_{\mathrm{tor}} \boldsymbol{u}^{\perp}, & \mathrm{if}\,\theta > 0 \\ -k_{\mathrm{tor}} \boldsymbol{u}^{\perp}, & \mathrm{if}\,\theta < 0 \end{cases} \qquad (5)$$

where
$k_{\mathrm{tor}}$ is the relative strength of $\boldsymbol{F}_{\mathrm{tor}}$. We set $k_{\mathrm{tor}} = 0.1$;
$\theta$ is the signed angle between $\boldsymbol{u}_r$ and $\boldsymbol{u}$; and
$\boldsymbol{u}^{\perp}$ is a unit vector rotated $90°$ counterclockwise relative to $\boldsymbol{u}$.

**Temporal forces** prevent slices from drifting too far from their original positions along the time axis positions (Fig. 7), which could cause unexpected accelerations and decelerations in the final animation. For every vertex, we compute the temporal force $\boldsymbol{F}_{\mathrm{tmp}}$ as

$$\boldsymbol{F}_{\mathrm{tmp}} = k_{\mathrm{tmp}} \boldsymbol{u}^{t} (t - t') \qquad (6)$$

where
$k_{\mathrm{tmp}}$ is the relative strength of $\boldsymbol{F}_{\mathrm{tmp}}$. We set $k_{\mathrm{tmp}} = 1$;
$t$ is the initial time of the slice to which the vertex belongs;
$t'$ is the current time value of the vertex; and
$\boldsymbol{u}^{t} = (0, 0, 1)$ .

**Computing total force and numerical integration:**
The total force on a vertex is the sum of all of the individual forces described above:

$$\boldsymbol{F}_{\mathrm{total}} = \boldsymbol{F}_{\mathrm{rpl}} + \boldsymbol{F}_{\mathrm{edg}} + \boldsymbol{F}_{\mathrm{bdr}} + \boldsymbol{F}_{\mathrm{ovr}} + \boldsymbol{F}_{\mathrm{tor}} + \boldsymbol{F}_{\mathrm{tmp}} \qquad (7)$$

We use explicit Euler integration to simulate the motions of the mesh vertices under the forces described above. Every vertex has a position and a velocity vector; in every iteration, we update velocities using forces, and update positions using velocities. These updates are scaled by a time step $\Delta t_{\mathrm{sim}}$ that we set to $0.01$. We cap velocities at $10\Delta t_{\mathrm{sim}}$ to dissipate extra energy from the simulation.

## 5.1 Spatial Queries

Repulsion and overlap forces rely on being able to find points on neighbouring elements that are close to a given query vertex. To find these points, we use each element's *envelope*, a triangle mesh implied by the construction in Sect. 3. Each triangle of the envelope is made from two time edges and one edge of a slice boundary, as shown in Fig. 8a. Given a query vertex $x$, we need to find nearby envelope triangles that belong to other elements.

To accelerate this computation, we first find and store the centroids of every element's envelope triangles in a uniformly subdivided 3D grid that surrounds the spacetime volume of the animation. In using this data structure, we make two simplifying assumptions; first, that because envelope triangles are small, their centroids are adequate for finding triangles near a given query point; and second, that the repulsion force from a more distant triangle is well approximated by a force from its centroid.

Given a query vertex $x$, we first find all envelope triangle centroids in nearby grid cells that belong to other elements. For each centroid, we use a method described by Ericson [12] to find the point on its triangle closest to $x$ and include that point in the list of points in Eq. (1). These nearby triangles will also be used to test for interpenetration of elements. We then find centroids in more distant grid cells, and add those centroids directly to the Eq. (1) list, skipping the closest point computation. In our system we set the cell size to $0.04$, giving a $25 \times 25 \times 25$ grid around the simulation volume. A query point's nearby grid cells are the 27 cells making up a $3 \times 3 \times 3$ block around the cell containing the point; the more distant cells are the 98 that make up the outer shell of the $5 \times 5 \times 5$ block around that (Fig. 8).

## 5.2 Slice Constraints

There are three hard geometric constraints on the configuration of slices, which must be enforced throughout the simulation. Each of the following constraints is reapplied after each physical simulation step described above.

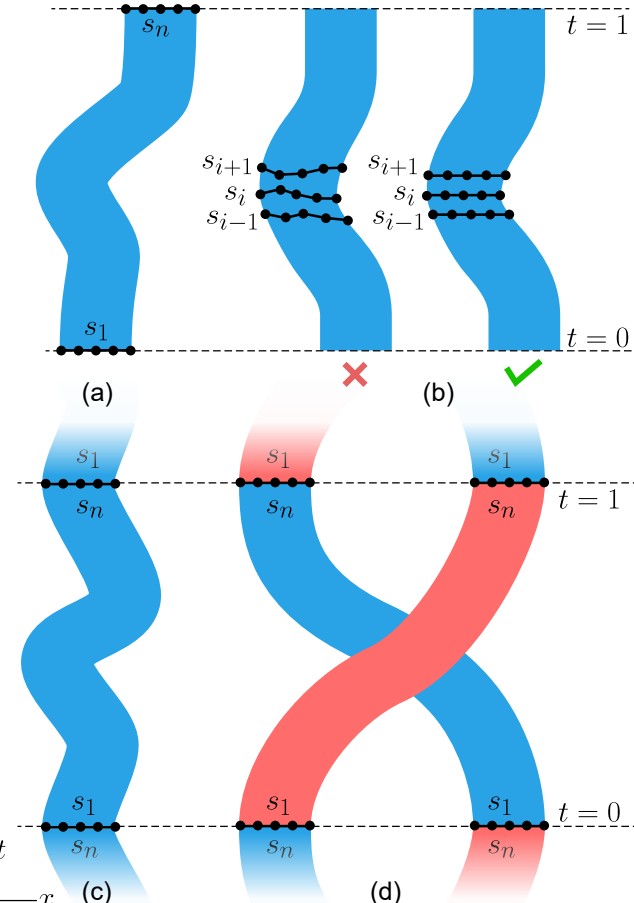

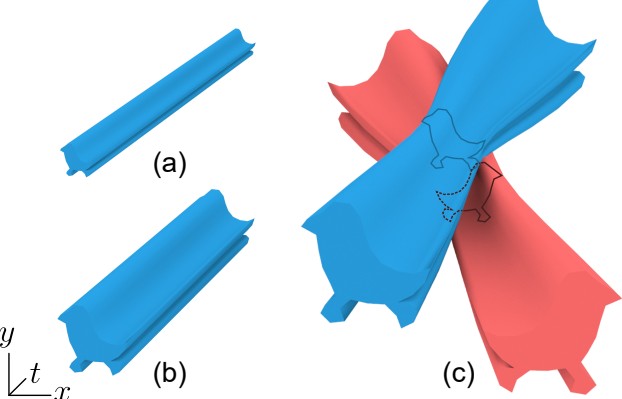

Figure 10: A spacetime element shown (a) shrunken at the beginning of the simulation, and (b) grown later in the simulation. (c) When two elements overlap somewhere along their lengths, they are temporarily prohibited from growing there.

$\boldsymbol{p}_2 = (x_2, y_2, 0) \in e_2$ are in correspondence, then after every simulation step we move $\boldsymbol{p}_1$ to $\left(\frac{x_1+x_2}{2}, \frac{y_1+y_2}{2}, 1\right)$ and $\boldsymbol{p}_2$ to $\left(\frac{x_2+x_1}{2}, \frac{y_2+y_2}{2}, 0\right)$.

### 5.3 Element Growth and Stopping Criteria

We begin the spacetime packing process with all element slices scaled down in $x$ and $y$, guaranteeing that elements do not overlap. As the simulation progresses we gradually grow the slices, consuming the negative space around them (Fig. 10a,b). A perfect packing would fill the spacetime container completely with the elements. Because each element wraps the underlying animated shape with a narrow channel of negative space, this would yield an even distribution of shapes in the resulting animation. For real-world elements, the goal of minimizing deformation of irregular element shapes will lead to imperfect packings with additional pockets of negative space.

**Element growth:** We induce elements to grow spatially by gradually increasing the rest lengths of their springs. The initial rest length of each spring is determined by the vertex positions in the shrunken version of the spacetime element constructed in Sect. 4. We allow an element's slices to grow independently of each other, which complicates the calculation of new rest lengths for time springs. Therefore, we create a duplicate of every shrunken spacetime element in the container, with a straight extrusion for unguided elements, and a polygonal extrusion for guided elements. This duplicate is not part of the simulation; it serves as a reference. Every element slice maintains a current scaling factor $g$. When we wish to grow the slice, we increase its $g$ value. We can compute new rest lengths for all springs by scaling every slice of the reference element by a factor of $g$ relative to the slice's centroid, and measuring distances between the scaled vertex positions. These new rest lengths are then used as the $\ell$ values in Equation 2.

Every element slice has its $g$ value initialized to 1. After every simulation step, if none of the slice's vertices were found to overlap other elements we increase that slice's $g$ by $0.001\Delta t_{\text{sim}}$, where $\Delta t_{\text{sim}}$ is the simulation time step. If any overlaps are found, then that slice's growth is instead paused to allow overlap and repulsion forces to give it more room to grow in later iterations. This approach can cause elements to fluctuate in size during the course of an animation, as slices compete for shifting negative space (Fig. 10).

**Stopping Criteria:** We halt the simulation when the space between neighbouring elements drops below a threshold. When calculating repulsion forces, we find the distance from every slice vertex to the closest point in a neighbouring element. The minimum of

---

Figure 9: a) End-to-end constraint: slice $s_1$ and $s_n$, located at $t = 0$ and $t = 1$, should never change their $t$ positions but can change their $x, y$ positions. b) Simultaneity constraint: all vertices on the same slice should have the same $t$ position. c) Loop constraint with a single element: the $x, y$ positions for $s_1$ and $s_n$ must match. d) Loop constraint with two elements: the $x, y$ position for $s_1$ for one element matches the $x, y$ position for $s_n$ of the other.

1. **End-to-end constraint:** A spacetime element must be present for the full length of the animation from $t = 0$ to $t = 1$. After every simulation step, every vertex belonging to an element's first slice has its $t$ value set to 0, and every vertex of the last slice has its $t$ value set to 1 (Fig. 9a).

2. **Simultaneity constraint:** During simulation, the vertices of a slice can drift away from each other in time, which could lead to rendering artifacts in the animation. After every simulation step, we compute the average $t$ value of all vertices belonging to each slice other than the first and last slices, and snap all the slice's vertices to that $t$ value (Fig. 9b).

3. **Loop constraint:** AnimationPak optionally supports looping animations. When looping is enabled, we must ensure that the $t = 0$ and $t = 1$ planes of the spacetime container are identical. The $t = 1$ slice of every element $e_1$ must then coincide with the $t = 0$ slice of *some* element $e_2$. We can have $e_1 = e_2$ (Fig. 9c), but more general loops are possible in which the elements arrive at a permutation of their original configuration (Fig. 9d). We require only that there is a one-to-one correspondence between the vertices of the $t = 1$ slice of $e_1$ and the $t = 0$ slice of $e_2$. If $\boldsymbol{p}_1 = (x_1, y_1, 1) \in e_1$ and

these distances over all vertices in an element slice determines that slice's closest distance to neighbouring elements. We halt the simulation when the maximum per-slice distance falls below 0.006 (relative to a normalized container size of 1). That is, we stop when every slice is touching (or nearly touching) at least one other element.

In some cases it can be useful to stop early based on cumulative element growth. In that case, we set a separate threshold for the slice scaling factors $g$ described above, and stop when the $g$ values of all slices exceed that threshold.

## 6 RENDERING

The result of the simulation described above is a packing of space-time elements within a spacetime container. We can render an animation frame-by-frame by cutting through this volume at evenly spaced $t$ values from $t = 0$ to $t = 1$. For our results, we typically render 500-frame animations.

During simulation, a given spacetime element's slices may drift from their original creation times. However, time springs keep the sequence monotonic, and the simultaneity constraint ensures that every slice is fixed to one $t$ value. To render this element at an arbitrary frame time $t_f \in [0, 1]$, we find the two consecutive slices whose time values bound the interval containing $t_f$ and linearly interpolate the vertex positions of the triangulations at those two slices to obtain a new triangulation at $t_f$. We can then compute a deformed copy of the original element paths by "replaying" the barycentric coordinates computed in Sect. 3 relative to the displaced triangulation vertices. We repeat this process for every spacetime element to obtain a rendering of the frame at $t_f$.

This interpolation process can occasionally lead to small artifacts in the animation. A rendered frame can fall between the discretely sampled slices for two elements at an intermediate time where physical forces were not computed explicitly. It is therefore possible for neighbouring elements to overlap briefly during such intervals.

## 7 IMPLEMENTATION AND RESULTS

The core AnimationPak algorithm consists of a C++ program that reads in text files describing the spacetime elements and the container, and outputs raster images of animation frames.

Large parts of AnimationPak can benefit from parallelism. In our implementation we update the cells of the collision grid (Sect. 5.1) in parallel by distributing them across a pool of threads. When the updated collision grid is ready, we distribute the spacetime elements over threads. We calculate forces, perform numerical integration, and apply the end-to-end and simultaneity constraints for each element in parallel. We must process any loop constraints afterwards, as they can affect vertices in two separate elements.

We created the results in this paper using a Windows PC with a 3.60GHz Intel i7-4790 processor and 16 GB of RAM. We used a pool of eight threads, corresponding to the number of logical CPU cores. Table 1 shows statistics for our results. Each packing has tens of thousands of vertices and hundreds of thousands of springs, and requires about an hour to complete. We enable the loop constraint in all results. The paper shows selected frames from the results; see the accompanying videos for full animations.

Fig. 1 is an animation of aquatic fauna featuring two penguins as guided elements. During one loop the penguins move clockwise around the container, swapping positions at the top and the bottom. Each ends at the other's starting point, demonstrating a loop constraint between distinct elements. All elements are animated, as shown in Fig. 1a. Note the coupling between the Pac-Man fish's mouth and the shark's tail on the left side of the second and fourth frames.

A snake chases a bird around an annular container in Fig. 11, demonstrating a container with a hole and giving a simple example of the narrative potential of animated packings. Fig. 12 animates the giraffe-to-penguin illusion shown as a static packing in Repulsion-Pak. This example uses torsional forces to control slice orientations.

Fig. 13 offers a direct comparison between packings computed using Centroidal Area Voronoi Diagrams (CAVD) [33], the spectral approach [10], and AnimationPak. These packings use stars that rotate and pulsate. For each method we show the initial frame ($t = 0$) and the halfway point ($t = 0.5$). The CAVD approach produces a satisfactory—albeit loosely coupled—packing for the first frame, but because the algorithm was not intended to work on animated elements, the evenness of the packing quickly degrades in later frames. The spectral approach is much better than CAVD, but their animated elements still have fixed spacetime shapes and can only translate and rotate to improve their fit. Repulsion forces and deformation allow AnimationPak to achieve a tighter packing that persists across the animation, including gear-like meshing of oppositely-rotating stars.

Fig. 14a is a static packing of a lion created by an artist and used as an example in FLOWPAK [32]. In Fig. 14b, we reproduce it with animated elements for the mane. The orientations of elements follow a vector field inside the container, and are maintained during the animation by torsional forces. We simulate only half of the packing and reflect it to create the other half. The facial features were added manually in a post-processing step.

Fig. 15 compares a static 2D packing created by RepulsionPak with a frame from an animated packing created by AnimationPak. The extra negative space in AnimationPak comes partly from the trade-off between temporal coherence and tight packing, and partly from the lack of secondary elements, which were used in a second pass in RepulsionPak to fill pockets of negative space.

Fig. 16 emphasizes the trade-off between temporal coherence and evenness of negative space by creating two animations with different time springs stiffness. In (a), the time springs are 100 times stronger than in (b). The resulting packing has larger pockets of negative space, but the accompanying video shows that the animation is smoother. The packing in (b) is tighter, but the elements must move frantically to maintain that tightness.

Fig. 17 is a failed attempt to animate a "blender". The packing has a beam that rotates clockwise and a number of small unguided circles. In a standard physics simulation we might expect the beam to push the circles around the container, giving each one a helical spacetime trajectory. Instead, as elements grow, repulsion forces cause circles to explore the container boundary, where they discover the lower-energy solution of slipping past the edge of the beam as it sweeps past. If we extend the beam to the full diameter of the container, consecutive slices simply teleport across the beam, hiding the moment of overlap in the brief time interval where physical forces were not computed. AnimationPak is not directly comparable to a 3D physics simulation; it is better suited to improving the packing quality of an animation that has already been blocked out at a high level.

## 8 CONCLUSION AND FUTURE WORK

We introduced AnimationPak, a system for generating animated packings by filling a static container with animated elements. Every animated 2D element is represented by an extruded spacetime tube. We discretize elements into triangle mesh slices connected by time edges, and deform element shapes and animations using a spacetime physical simulation. The result is a temporally coherent 2D animation of elements that attempt both to perform their scripted motions and consume the negative space of the container. We show a variety of results where 2D elements move around inside the container.

We see an number of opportunities for improvements and extensions to AnimationPak:

- Because we use linear interpolation to synthesize an element's shape between slices, we require elements not to undergo

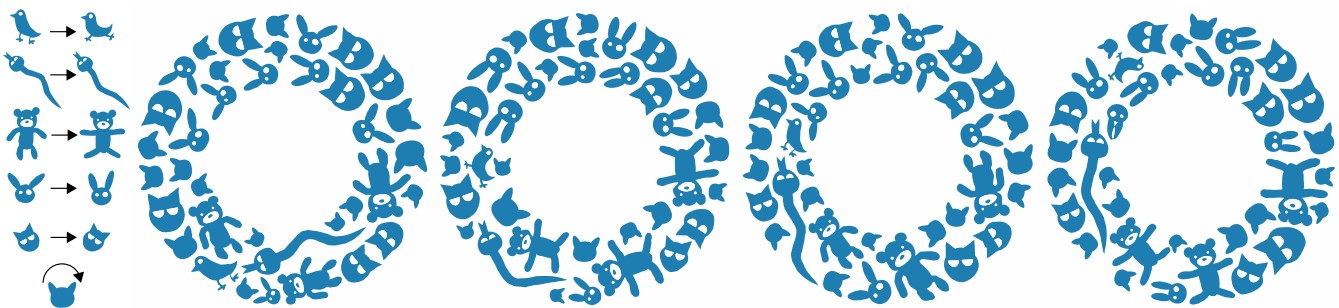

Figure 11: A snake chasing a bird through a packing of animals. The snake and bird are both guided elements that move clockwise around the annular container.

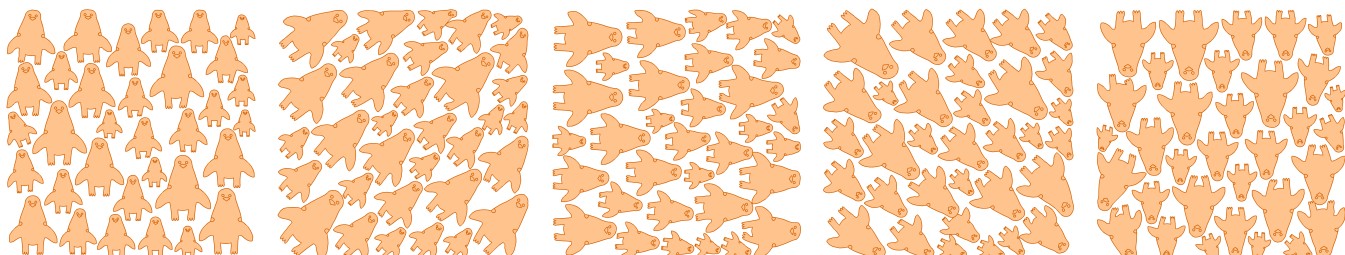

Figure 12: Penguins turning into giraffes. The penguins animate by rotating in place. Torsional forces are used to preserve element orientations. Frames are taken at $t = 0$, $t = 0.125$, $t = 0.25$, $t = 0.375$, and $t = 0.5$.

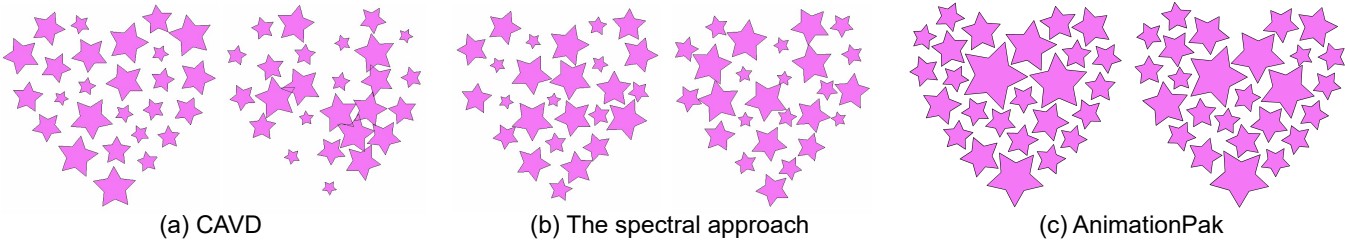

(a) CAVD        (b) The spectral approach        (c) AnimationPak

Figure 13: A comparison of (a) Centroidal Area Voronoi Diagrams (CAVDs) [33], (b) spectral packing [10], and (c) AnimationPak. We show two frames for each method, taken at $t = 0$ and $t = 0.5$. The CAVD packing starts with evenly distributed elements but the packing degrades as the animation progresses. The spectral approach improves upon CAVD with better consistency, but still leaves significant pockets of negative space. The AnimationPak packing has less negative space that is more even.

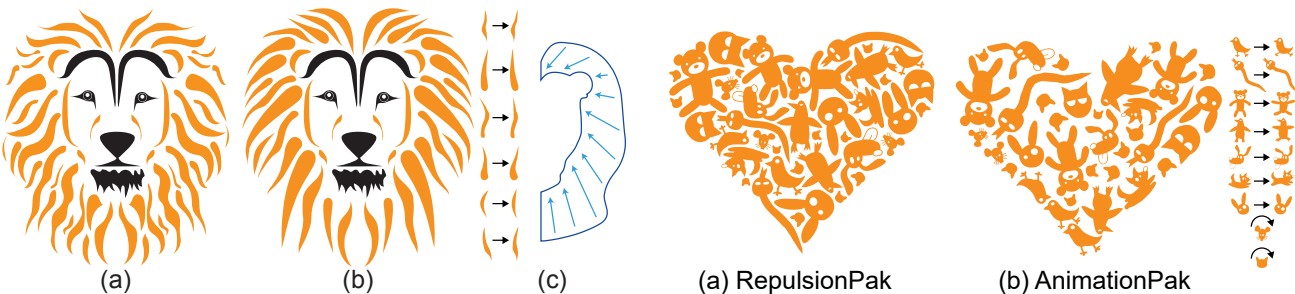

(a)        (b)        (c)        (a) RepulsionPak        (b) AnimationPak

Figure 14: (a) A static packing made by an artist, taken from StockUnlimited. (b) The first frame from an AnimationPak packing. (c) The input animated elements and the container shape with a vector field. Torsional forces keep elements oriented in the direction of the vector field. We simulate half of the lion's mane and render the other half using a reflection, and add the facial features by hand.

Figure 15: (a) A static packing created with RepulsionPak. (b) The first frame of a comparable AnimationPak packing. The input spacetime elements are shown on the right. The AnimationPak packing has more negative space because we must tradeoff between temporal coherence and packing density.

Table 1: Data and statistics for the results in the paper. The table shows the number of elements, the number of vertices, the number of springs, the number of envelope triangles, and the running time of the simulation in hours, minutes, and seconds.

| Packing | Elements | Vertices | Springs | Triangles | Time |
|---|---|---|---|---|---|
| Aquatic animals (Fig. 1) | 37 | 97,800 | 623,634 | 106,000 | 01:06:35 |
| Snake and birb (Fig. 11) | 37 | 58,700 | 370,571 | 58,700 | 01:01:32 |
| Penguin to giraffe (Fig. 12) | 33 | 124,300 | 824,164 | 143,000 | 01:19:50 |
| Heart stars (Fig. 13c) | 26 | 85,200 | 598,218 | 858,00 | 00:23:08 |
| Animals (Fig. 15b) | 34 | 69,600 | 444,337 | 69,800 | 01:00:19 |
| Lion (Fig. 14b) | 16 | 39,400 | 236,086 | 41,800 | 00:41:56 |

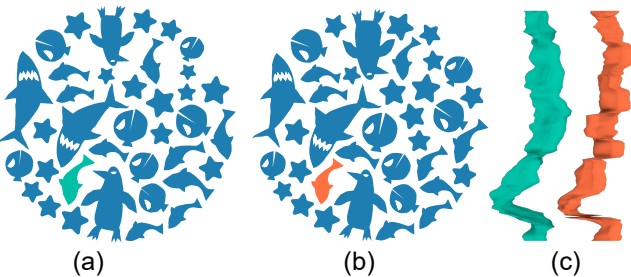

(a)     (b)     (c)

Figure 16: (a) One frame from Fig. 1. (b) The same packing with time springs that are 1% as stiff. Reducing the stiffness of time springs leads to a more even packing with less negative space, but the animated elements must move frantically to preserve packing density. The spacetime trajectories of the highlighted fish in (a) and (b) are shown in (c). The orange fish in (b) exhibits more high frequency fluctuation in its position.

changes in topology. More sophisticated representations of vector shapes, such as that of Dalstein et al. [11], could support interpolations between slices with complex topological changes. We would also need to synthesize a watertight envelope around the animating element in order to compute overlap and repulsion forces.

- We would like to improve the performance of the physical simulation. One option may be to increase the resolution of element meshes progressively during simulation. Early in the process, elements are small and distant from each other, so lower-resolution meshes may suffice for computing repulsion forces.

- As noted in Sect. 6 and Fig. 17, our discrete simulation can miss element overlaps that occur between slices. A more robust continuous collision detection (CCD) algorithm such as that of Brochu et al. [3] could help us find all collisions between the envelopes of spacetime elements.

- In RepulsionPak [30], an additional pass with small secondary elements had a significant positive effect on the distribution of negative space in the final packing. It may be possible to identify stretches of unused spacetime that can be filled opportunistically with additional elements. The challenge would be to locate tubes of empty space that run the full duration of the animation, always of sufficient diameter to accommodate an added element.

- Like the spectral method [10], and unlike Animosaics [33], AnimationPak can pack animated elements into a static con-

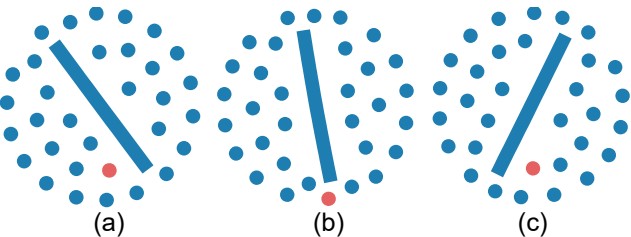

(a)     (b)     (c)

Figure 17: A failure case for AnimationPak, consisting of a rotating beam and a number of small circles. Instead of being dragged around by the beam, the circles dodge it entirely by sneaking through the gap between the beam and the container. The red circle demonstrates one such maneuver.

tainer. We would like to extend our work to also handle animated containers. This extension would certainly affect the initial element placement, which would need to ensure that elements are placed fully inside the spacetime volume of the container. It could also lead to undesirable scaling of elements if the container area changes too much. It would be interesting to investigate whether we could adapt to changes in area by adding and removing elements unobtrusively during the animation, in the style of Animosaics.

- AnimationPak implements forces and constraints geared towards spacetime animation, but many of the same ideas could be adapted to develop a deformation-driven method for packing purely spatial 3D objects into a 3D container. We would like to evaluate the expressivity and visual quality of deformation-driven 3D packings in comparison to other 3D packing techniques.

- Our physical simulation relies in several places on our method of constructing and animating spacetime elements. Our time edges make use of the one-to-one correspondence between boundary vertices of adjacent slices in order to construct a mesh surface that bounds each element. We also make direct use of that correspondence when rendering, to interpolate new triangulations between existing slices. We would like AnimationPak to be more agnostic about the method used to create animated elements. Given a "generic" animated element, we can easily compute independent triangulated slices, but we would need robust algorithms to join them into an extrusion and interpolate within that extrusion later.

- Saputra et al. [30] previously studied a set of measurements inspired by spatial statistics for evaluating the evenness of the distribution of negative space in a static packing. While their measurements extend naturally to three purely spatial dimensions, it is not clear whether they can be adapted to our spacetime context. We would like to investigate spatial statistics for the quality of animated packings that correlate with human perceptual judgments.

- There are many examples of static two-dimensional packings created by artists, which can serve as inspiration for an algorithm like RepulsionPak. We were unable to find an equivalent set of animated examples, probably because they would be difficult and time-consuming to create by hand. We would like to engage with artists to understand the aesthetic value and limitations of AnimationPak.

## ACKNOWLEDGMENTS

We thank reviewers for helpful feedback. Thanks to Danny Kaufman for discussions about spacetime optimizations and physics simu-

lations. This research was funded by the National Sciences and Engineering Research Council of Canada (NSERC) and through a generous gift from Adobe.

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
