# OpenReview forum: "AnimationPak: Packing Elements with Scripted Animations"
_graphicsinterface.org/Graphics_Interface/2020/Conference — GI 2020_

### Official Review · AnonReviewer3 · 2020-04-18
**Quality needs improvement**

**Rating:** 4
**Confidence:** 4

**Review:**

This paper proposes a follow-up idea to RepulsionPak, extending the static packing to a dynamic packing animation.

The idea of treating the time domain as the 3rd dimension in physical simulation is neat. Similar ideas have been studied in space-time optimization.

Watching the video, I think the animation quality needs more improvement. Most of my concerns are around quality.

1) The generation of the guided elements is to interpolate between a small set of target positions. Since the driving animations from these guided elements are simple, the final movement of all the shapes is not very dynamic.

2) Watching Fig1 video and Fig13 video, it seems like to perturb a few key shapes and everything else follows the repulsion force. The most interesting video to me is Fig17 since there is consistent and large motion. All other videos seem like random perturbation.

3) I don’t know if this is out of scope for this paper, but I do find it hard to define “What is a good AnimationPak?”.  The motivating example from Unilever is static. I tried to search online to find a baseline on what is a good animation packing for abstract shapes and it’s unclear. I found this video to animate the Unilever logo, but it’s not 2D packing, as described in this paper. https://www.youtube.com/watch?v=km7iP8V6Ytw

4) Animation packing is new and it’s probably hard to find direct comparisons with previous work. One possibility is to run a 2D simulation in the time domain as opposed to the current 3D space-time formulation. For Fig11 snake_birb example, I am curious how it would perform with a 2D simulator.

---

### Official Review · AnonReviewer1 · 2020-04-20
**nice work, should accept**

**Rating:** 7
**Confidence:** 4

**Review:**

This paper extends the work of Saputra et al (GI 2018, TVCG 2019) on static packing to animated packings where each element potentially has some sort of scripted behaviour. The method creates a static optimization over a timespace cube, which then can be interpreted as an 2D animation by taking cross-sections perpendicular to the cube's time dimension.
It is quite well written and has worthwhile results. It is a clear accept, somewhere between 7 and 8 on OpenReview's 10-point scale.

The paper does an excellent job of describing the method in detail. In one respect, though, more discussion would be warranted. I am not sure about the directability of these animations -- i.e., how the scripting is meant to be done. There are tantalizing statements such as "The artist can optionally specify trajectories for a subset of the elements". How are these trajectories specified, and how time-consuming is it to specify constraints? I understand that the authors have not built an animation tool. Still, it would be nice if the paper gave some indication of the amount of human effort needed to create the animations shown. The animations are mostly not very sophisticated, and I wondered what could be done with them. Some examples, such as the lion's mane animation, make good use of the limited movements.

The technique works nicely and the animations seem decent. It is hard to judge success in work like this with no clear antecedent. Controlling smoothness, probably with higher-order continuity of the spacetime worms, will be an obvious direction for future work. The paper shows a variety of examples demonstrating some of the possible uses. In their closing remarks, the authors suggest that artists will find additional imaginative uses for this work. I concur. The paper addresses the problem posed but also opens the door to future unknown possibilities.

Minor points:

Figure 15 compares RepulsionPak and AnimationPak, and judges RepulsionPak to have higher "packing quality". RepulsionPak packs elements more tightly, which I suppose is what is meant by "quality". It might be better to use a less judgemental term and say something like "packing density".

The authors might like to compare with the abstract animations of gMotion, GI 2018.

---

### Official Review · AnonReviewer2 · 2020-04-24
**Straightforward, but interesting nevertheless**

**Rating:** 7
**Confidence:** 4

**Review:**

The paper presents a new system for a new application: packing animated elements within a given shape ('container'). The overall idea of the system is to model this as a physics-inspired system. Given shape elements are first picked randomly from the collection, their centroids distributed as a blue noise over the container shape. Then the elements repulse, deform each other, move etc.; at each time step the system models the necessary forces, and the final animation is the time integration of that. In addition, to prevent overlaps, the elements are initialized small, and are gradually grown, whenever possible.

The final results look quite appealing. Even though I can not think of any application beyond pure graphics, I don't mind. The text is written clearly, and even though it often answers the question 'what we do' instead 'why', the 'why' is almost everywhere easy to decipher.

I have a few reservations about the paper:
	- I don't really understand the value of introducing spacetime here. I do understand that it's a convenient visualization or another point of view that this process can be viewed as packing a spacetime, but I don't think the authors really use it beyond pure visualization. I would appreciate if the authors could clarify that in the text.
	- While I do understand that this is a new and purely graphics-related application, the quickly varying size of elements during the animation seems... suboptimal. I would have expected the animation to be as-rigid-as-possible. I guess it would make the problem more complex, but I would nevertheless appreciate some justification in the text.
	- Also a clarification point: the repulsion force seems to care about the vertices on the convex hull of each element, but in the final results it seems like the elements don't intersect, but their convex hulls do (which is great, that's what I'd expect). So it is because the weight of the repulsion force is small, so it's mildly unhappy?
	- This is very reminiscent of the spacetime in 'Vector Graphics Complexes' by Dalstein et al. (SIGGRAPH 2014). I think it would be a good addition to the related work.

A few minor comments:
	- While I can guess the electric repulsion inspiration in the repulsion force, edge force is a little surprising. Why is it quadratic? Which physical model of elasticity is it inspired by?
	- L89: CAVDs -- abbreviation without explanation
	- 3.2 seems a little bizarre from a UI standpoint: why isn't it possible to just take an existing animation in standard format?

Other than that, I think the results look nice, the technique makes sense; the contribution, albeit modest, is still quite interesting.

---

### Meta-Review · Area_Chair1 · 2020-04-25

**Recommendation:** Accept
**Confidence:** 3

**Metareview:**

This paper presents a method for animating 2D shapes packed into a container space. Reviews were split, with two of three recommending acceptance. The majority view saw the work as solid and clearly explained. Reviewers raised concerns about animation quality and control, and revisions could expand the discussion on these issues, space permitting. Some design decisions could be further explained.


Pros:

- clear improvement over past results on animation packings
- thorough explanation of technical aspects

Cons:

- animations are not convincing to all viewers
- unclear to easy it is to control the animations; possibly linked to above issue
- some design decisions lack clear motivation

---

### Decision · Program_Chairs · 2020-04-25

Accept